# ALIM: Adjusting Label Importance Mechanism for Noisy Partial Label Learning

**Mingyu Xu**[1,2*], **Zheng Lian**[1*], **Lei Feng**[3], **Bin Liu**[1,2], **Jianhua Tao**[4,5]
[1]The State Key Laboratory of Multimodal Artificial Intelligence Systems,
Institute of Automation, Chinese Academy of Sciences
[2]School of Artificial Intelligence, University of Chinese Academy of Sciences
[3]School of Computer Science and Engineering, Nanyang Technological University
[4]Department of Automation, Tsinghua University
[5]Beijing National Research Center for Information Science and Technology, Tsinghua University
{xumingyu2021, lianzheng2016}@ia.ac.cn

## Abstract

Noisy partial label learning (noisy PLL) is an important branch of weakly supervised learning. Unlike PLL where the ground-truth label must conceal in the candidate label set, noisy PLL relaxes this constraint and allows the ground-truth label may not be in the candidate label set. To address this challenging problem, most of the existing works attempt to detect noisy samples and estimate the ground-truth label for each noisy sample. However, detection errors are unavoidable. These errors can accumulate during training and continuously affect model optimization. To this end, we propose a novel framework for noisy PLL with theoretical interpretations, called "Adjusting Label Importance Mechanism (ALIM)". It aims to reduce the negative impact of detection errors by trading off the initial candidate set and model outputs. ALIM is a plug-in strategy that can be integrated with existing PLL approaches. Experimental results on multiple benchmark datasets demonstrate that our method can achieve state-of-the-art performance on noisy PLL. Our code is available at: https://github.com/zeroQiaoba/ALIM.

## 1 Introduction

Partial label learning (PLL) [1, 2] (also called ambiguous label learning [3, 4] and superset label learning [5, 6]) is a typical type of weakly supervised learning. Unlike supervised learning where each sample is associated with a ground-truth label, PLL needs to identify the ground-truth label from a set of candidate labels. Due to the low annotation cost of partially labeled samples, PLL has attracted increasing attention from researchers and has been applied to many areas, such as object recognition [4], web mining [7], and ecological informatics [8].

The basic assumption of PLL is that the ground-truth label must be in the candidate label set [9]. However, this assumption may not be satisfied due to the unprofessional judgment of annotators [10]. Recently, some researchers have relaxed this assumption and focused on a more practical setup called noisy PLL [11]. In noisy PLL, the ground-truth label may not be in the candidate label set. To deal with this task, Lv et al. [11] utilized noise-tolerant loss functions to avoid overemphasizing noisy samples. However, they cannot fully exploit the useful information in noisy data. To this end, Lian et al. [12] and Wang et al. [13] proposed to detect noisy samples and estimate pseudo labels for these samples. However, detection errors are unavoidable. These errors will accumulate and continuously affect model optimization, thereby limiting their performance on noisy PLL.

---

[*]Equal Contribution

37th Conference on Neural Information Processing Systems (NeurIPS 2023).

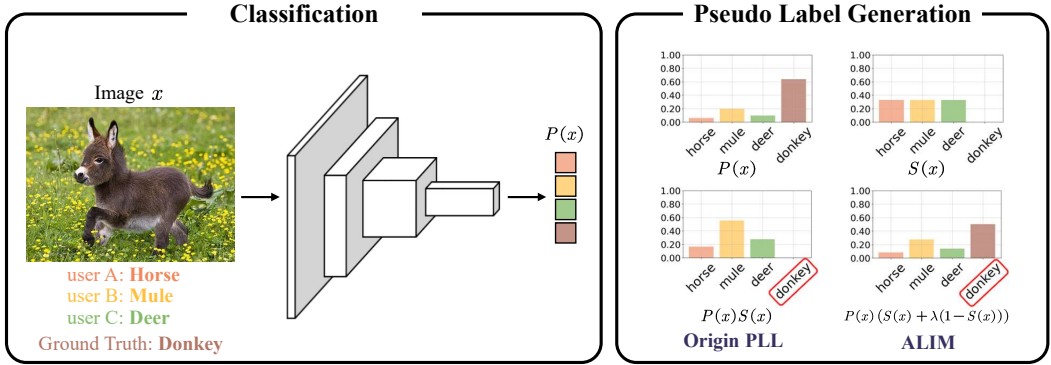

Figure 1: The core structure of ALIM. The network receives an input $x$ and produces softmax prediction probabilities $P(x)$. Different from traditional PLL that fully trusts the candidate set, our method can deal with noisy samples through a weighting mechanism.

To alleviate this problem, we propose a novel framework for noisy PLL called "Adjusting Label Importance Mechanism (ALIM)". Although we may make mistakes in noisy sample detection, it allows us to leverage the initial candidate set and restart correction. To reduce manual efforts in hyper-parameter tuning, we propose an adaptive strategy to determine the weighting coefficient for ALIM. To further improve noise tolerance, we equip ALIM with mixup training [14], a powerful technique in noisy label learning [15]. We also perform theoretical analysis from the perspective of objective functions and EM algorithms and prove the feasibility of our method. To verify its effectiveness, we conduct experiments on multiple benchmark datasets. Experimental results demonstrate that ALIM outperforms currently advanced approaches, setting new state-of-the-art records. The main contributions of this paper can be summarized as follows:

- We propose a plug-in framework for noisy PLL with theoretical interpretations. Our method can deal with noisy samples by trading off the initial candidate set and model outputs.

- We propose an adaptive strategy to adjust the weighting coefficient for ALIM. Furthermore, we combine our method with the mixup training to improve noise robustness.

- Experimental results on multiple datasets demonstrate the effectiveness of our method. ALIM is superior to currently advanced approaches on noisy PLL.

## 2 Methodology

### 2.1 Problem Definition

Let $\mathcal{X}$ be the input space and $\mathcal{Y}$ be the label space with $c$ distinct categories. We consider a dataset $\mathcal{D}$ containing $N$ partially labeled samples $\{(x, S(x))\}$. Here, $S(x) \in \{0, 1\}^c$ is the candidate set for the sample $x \in \mathcal{X}$. We denote the $i^{th}$ element of $S(x)$ as $S_i(x)$, which is equal to 1 if the label $i$ is in the candidate set, and 0 otherwise. Meanwhile, we denote $P(x) \in \mathbb{R}^c$, $w(x) \in \mathbb{R}^c$, and $y(x) \in \mathbb{R}^1$ as the prediction probabilities, the pseudo label, and the ground-truth label of the sample $x$. The $i^{th}$ elements of $P(x)$ and $w(x)$ are represented as $P_i(x)$ and $w_i(x)$, respectively. In this paper, all product operations between matrices are Hadamard products.

The goal of noisy PLL is to learn a classifier that can minimize the classification risk on $\mathcal{D}$. Unlike PLL, noisy PLL allows the ground-truth label may not be in the candidate set. For a fair comparison, we adopt the same data generation procedure as previous works [12]. Specifically, to generate candidate labels, we first flip incorrect labels into false positive labels with a probability $q$ and aggregate the flipped ones with the ground-truth label. Then, we assume that each sample has a probability $\eta$ of being the noisy sample. For each noisy sample, we further select a label from the non-candidate set, move it into the candidate set, and move the ground-truth label out of the candidate set. We denote the probability $q$ as the ambiguity level and the probability $\eta$ as the noise level.

## 2.2 Motivation

First, let us review our own exam experience. When we are unfamiliar with a test, we believe that the correct answer must be in the candidate set. Even if every option is wrong, we still choose the most likely answer. But as we become more familiar with the test, we learn to question the correctness of the candidate set. If we believe every option is wrong, we will consider answers outside the candidate set. In noisy PLL, this strategy can also be used to handle noisy samples.

## 2.3 ALIM Framework

Inspired by the above idea, we propose a simple yet effective framework called ALIM. The overall structure is shown in Figure 1. Specifically, we first predict the softmax probabilities $P(x)$ for each sample $x$. In traditional PLL, we fully trust the candidate set and generate the pseudo label as follows:

$$w(x) = \text{Normalize}\left(S(x)P(x)\right), \tag{1}$$

where $\text{Normalize}(\cdot)$ is a normalization function that ensures $\sum_{i=1}^{c} w_i(x) = 1$. Different from traditional PLL, ALIM introduces a coefficient $\lambda$ to control the reliability of the candidate set:

$$\tilde{S}(x) = S(x) + \lambda\left(1 - S(x)\right), \tag{2}$$

$$w(x) = \text{Normalize}\left(\tilde{S}(x)P(x)\right). \tag{3}$$

Here, $\lambda = 0$ means that we fully trust the given candidate set $S(x)$; $\lambda = 1$ means that we don't believe the candidate set but trust our own judgment $P(x)$. According to Eq. 2, the time complexity of this process is mere $O(cN)$, where $c$ is the number of categories and $N$ is the number of samples.

In this paper, we discuss two choices of normalization functions: $\text{Onehot}(\cdot)$ and $\text{Scale}(\cdot)$. Specifically, $\text{Onehot}(\cdot)$ sets the maximum value to 1 and others to 0; $\text{Scale}(\cdot)$ introduces a scaling factor $K > 0$ and normalizes the probabilities as follows:

$$\text{Scale}(z) = \left\{\frac{z_i^{1/K}}{\sum_j z_j^{1/K}}\right\}_{i=1}^{c}, \tag{4}$$

where $z_i$ is the $i^{th}$ element of $z$.

## 2.4 Theoretical Analysis

To verify the feasibility of our method, we conduct theoretical analysis from two perspectives: (1) the manually-designed objective function; (2) the classic expectation-maximization (EM) algorithm [16].

### 2.4.1 Interpretation from Objective Functions

Let $P_i$, $S_i$, and $w_i$ be abbreviations for $P_i(x)$, $S_i(x)$, and $w_i(x)$, respectively. During training, we should optimize the following objectives:

- Minimize the classification loss on $w(x)$ and $P(x)$.
- $w(x)$ should be small at non-candidate labels.
- Entropy regularization on $w(x)$ to avoid overconfidence of pseudo labels.
- $w(x)$ should satisfy $0 \leq w_i \leq 1$ and $\sum_{i=1}^{c} w_i = 1$.

Then, the final objective function is calculated as follows:

$$\max \sum_{i=1}^{c} w_i \log P_i + M\left(\sum_{i=1}^{c} w_i S_i - 1\right) - K\sum_{i=1}^{c} w_i \log w_i$$

$$s.t. \sum_{i}^{c} w_i = 1, w_i \geq 0, \tag{5}$$

where $M$ and $K$ are penalty factors. By using Lagrange multipliers, we can observe that the penalty factor $K$ is different for two normalization functions: $K = 0$ for $\text{Onehot}(\cdot)$ and $K > 0$ for $\text{Scale}(\cdot)$. The penalty factor $M$ has a strong correlation with the weighting coefficient $\lambda$ in Eq. 2, i.e., $\lambda = e^{-M}$. Larger $M$ (or smaller $\lambda$) means that we have tighter constraints on $(\sum_{i=1}^{c} w_i S_i - 1)$, and therefore we should trust the given candidate set more. It is identical to the meaning of $\lambda$ in our ALIM (see Section 2.3).

### 2.4.2 Interpretation from EM Perspective

EM aims to maximize the likelihood of the corpus $\mathcal{D}$. Following previous works [5, 17], we first make a mild assumption:

**Assumption 1** *In noisy PLL, the ground-truth label may not be in the candidate set $S(x)$. We assume that each candidate label $\{i|S_i(x) = 1\}$ has an equal probability $\alpha(x)$ of generating $S(x)$ and each non-candidate label $\{i|S_i(x) = 0\}$ has an equal probability $\beta(x)$ of generating $S(x)$.*

Besides the interpretation from objective functions, we further explain ALIM from the EM perspectivein Appendix B. We prove that the E-step aims to predict the ground-truth label for each sample and the M-step aims to minimize the classification loss. Meanwhile, ALIM is a simplified version of the results derived from EM. Specifically, EM uses an instance-dependent $\lambda(x) = \beta(x)/\alpha(x)$, while ALIM uses a global $\lambda$. In the future, we will explore the performance of the instance-dependent $\lambda(x)$ for noisy PLL. Additionally, EM connects traditional PLL with noisy PLL. In traditional PLL, we assume that the ground-truth label must be in the candidate set, i.e., $\beta(x) = 0$. Then, our noisy PLL approach ALIM will degenerate to the classic PLL method RC [18].

## 2.5 Optional Key Components

We further introduce several key components to make our method more effective, including the adaptively adjusted strategy and the mixup training.

### 2.5.1 Adaptively Adjusted Strategy

Appropriate $\lambda$ is important for ALIM. Too small $\lambda$ makes us fully trust the candidate set, thus easily over-fitting on noise samples; too large $\lambda$ makes us ignore the prior information in the candidate set, thus limiting the classification performance. Therefore, we propose a manually adjusted strategy to find a proper $\lambda$ in a predefined parameter space.

To further reduce manual efforts, we also propose an adaptively adjusted strategy. Specifically, we first estimate the noise level of the dataset. Intuitively, we can randomly select a subset from $\mathcal{D}$, annotate the ground-truth labels by professional annotators, and estimate the noise level of the dataset. Alternatively, we can automatically estimate noise rate via the Gaussian mixture model [19, 20] or cross-validation [21, 22]. The estimated noise level is represented as $\eta$. Based on Appendix C, we prove that the value of Eq. 6 can be viewed as a metric, and the $\eta$-quantile of this value can be treated as the adaptively adjusted $\lambda$.

$$\left\{ \frac{\max_i S_i(x) P_i(x)}{\max_i (1 - S_i(x)) P_i(x)} \right\}_{x \in \mathcal{D}}. \tag{6}$$

In Section 3.2, We further present results without noise rate estimation and manually adjust $\lambda$ as a hyper-parameter. Through experimental analysis, this approach can also achieve competitive performance. Therefore, the adaptively adjusted strategy is optional. Its main advantage is to reduce manual efforts in hyper-parameter tuning and realize a more automatic approach for noisy PLL.

### 2.5.2 Mixup Training

Since the mixup training is powerful in noisy label learning [14, 15], we further combine this strategy with ALIM for noisy PLL. Consider a pair of samples $x_i$ and $x_j$ whose pseudo labels are denoted as $w(x_i)$ and $w(x_j)$, respectively. Then, we create a virtual training sample by linear interpolation:

$$x_{\text{mix}} = \alpha x_i + (1 - \alpha) x_j, \tag{7}$$

$$w_{\text{mix}} = \alpha w(x_i) + (1 - \alpha) w(x_j). \tag{8}$$

**Algorithm 1:** Pseudo-code of ALIM.

---

**Input:** Dataset $\mathcal{D}$ with the estimated noise level $\eta$, predictive model $f$, warm-up epoch $e_0$, number of epoch $E_{\max}$, weighting coefficient $\lambda$, trade-off between losses $\lambda_{\mix}$.

**Output:** The optimized model $f$.

**1** **for** $e = 1, \cdots, e_0$ **do**
**2** | Warm up by training $f$ with $\mathcal{L}_{\text{pll}}$;
**3** **end**

**4**

**5** **for** $e = e_0, \cdots, E_{max}$ **do**
**6** | **for** $\{x_i, S(x_i)\} \in \mathcal{D}$ **do**
**7** | | Calculate the output of the predictive model $f(x_i)$;
**8** | | Obtain the pseudo label $w(x_i)$ by Eq. 2∼3;
**9** | | Get the PLL loss $\mathcal{L}_{\text{pll}}$ between $f(x_i)$ and $w(x_i)$;
**10** | | **if** *mixup training* **then**
**11** | | | Sample $\alpha \sim \text{Beta}(\zeta, \zeta)$ and sample $x_j$ from $\mathcal{D}$;
**12** | | | Create the virtual sample $(x_{\mix}, w_{\mix})$ by Eq. 7∼8;
**13** | | | Calculate the mixup loss $\mathcal{L}_{\mix}$ between $f(x_{\mix})$ and $w_{\mix}$;
**14** | | | Get the final loss $\mathcal{L} = \mathcal{L}_{\text{pll}} + \lambda_{\mix}\mathcal{L}_{\mix}$;
**15** | | **else**
**16** | | | Get the final loss $\mathcal{L} = \mathcal{L}_{\text{pll}}$;
**17** | | **end**
**18** | **end**
**19** | Optimize $f$ by minimizing $\mathcal{L}$;

**20**

**21** | **if** *adaptively adjusted* $\lambda$ **then**
**22** | | Create an empty list $G$;
**23** | | **for** $\{x_i, S(x_i)\} \in \mathcal{D}$ **do**
**24** | | | Calculate the output of the predictive model $f(x_i)$;
**25** | | | Store the value in Eq. 6 to $G$;
**26** | | **end**
**27** | | $\lambda \leftarrow \eta-$quantile of the list $G$;
**28** | **end**
**29** **end**

---

$\alpha \sim \text{Beta}(\zeta, \zeta)$, where $\zeta$ is a parameter in the beta distribution. We define the mixup objective $\mathcal{L}_{\mix}$ as the cross-entropy loss on $P(x_{\mix})$ and $w_{\mix}$. During training, we combine the mixup loss $\mathcal{L}_{\mix}$ and the PLL loss $\mathcal{L}_{\text{pll}}$ into a joint objective function $\mathcal{L} = \mathcal{L}_{\text{pll}} + \lambda_{\mix}\mathcal{L}_{\mix}$, where $\lambda_{\mix}$ controls the trade-off between two losses. The pseudo-code of ALIM is summarized in Algorithm 1.

## 3 Experiments

### 3.1 Experimental Setup

**Corpus Description**   In the main experiments, we evaluate the performance on two benchmark datasets of noisy PLL, CIFAR-10 [23] and CIFAR-100 [23]. We choose the noise level $\eta$ from $\{0.1, 0.2, 0.3\}$. Since CIFAR-100 has more classes than CIFAR-10, we consider $q \in \{0.1, 0.3, 0.5\}$ for CIFAR-10 and $q \in \{0.01, 0.03, 0.05\}$ for CIFAR-100. In Section 3.2, we also conduct experiments on fine-grained datasets (CUB-200 [17] and CIFAR-100H [24]) and consider severe noise.

**Baselines**   To verify the effectiveness of our method, we implement the following state-of-the-art methods as baselines: 1) CC [18]: a classifier-consistent PLL method under the uniform candidate label generation assumption; 2) RC [18]: a risk-consistent PLL method under the same assumption as CC; 3) LWC [25]: a PLL method that considers the trade-off between losses on candidate and non-candidate sets; 4) LWS [25]: a PLL method that combines the weighted loss with the sigmoid activation function; 5) PiCO [17]: a PLL method using contrastive learning; 6) CRDPLL [26]: a PLL method that exploits consistency regularization on the candidate set and supervised learning on the non-candidate set; 7) IRNet [12]: a noisy PLL method that progressively purifies noisy samples; 8) PiCO+ [13]: a noisy PLL method using a semi-supervised contrastive framework.

Table 1: Performance of different methods. $\diamond$ denotes the models without mixup training, and $\heartsuit$ denotes the models with mixup training. By default, we combine ALIM with PiCO for noisy PLL.

| CIFAR-10 | $q=0.1$ | | | $q=0.3$ | | | $q=0.5$ | | |
|---|---|---|---|---|---|---|---|---|---|
| | $\eta=0.1$ | $\eta=0.2$ | $\eta=0.3$ | $\eta=0.1$ | $\eta=0.2$ | $\eta=0.3$ | $\eta=0.1$ | $\eta=0.2$ | $\eta=0.3$ |
| $\diamond$CC | 79.81±0.22 | 77.06±0.18 | 73.87±0.31 | 74.09±0.60 | 71.43±0.56 | 68.08±1.12 | 69.87±0.94 | 59.35±0.22 | 48.93±0.52 |
| $\diamond$RC | 80.87±0.30 | 78.22±0.23 | 75.24±0.17 | 79.69±0.37 | 75.69±0.63 | 71.01±0.54 | 72.46±1.51 | 59.72±0.42 | 49.74±0.70 |
| $\diamond$LWC | 79.13±0.53 | 76.15±0.46 | 74.17±0.48 | 77.47±0.56 | 74.02±0.35 | 69.10±0.59 | 70.59±1.34 | 57.42±1.14 | 48.93±0.37 |
| $\diamond$LWS | 82.97±0.24 | 79.46±0.09 | 74.28±0.79 | 80.93±0.28 | 76.07±0.38 | 69.70±0.72 | 70.41±2.68 | 58.26±0.28 | 39.42±3.09 |
| $\diamond$PiCO | 90.78±0.24 | 87.27±0.11 | 84.96±0.12 | 89.71±0.18 | 85.78±0.23 | 82.25±0.32 | 88.11±0.29 | 82.41±0.30 | 68.75±2.62 |
| $\diamond$CRDPLL | 93.48±0.17 | 89.13±0.39 | 86.19±0.48 | 92.73±0.19 | 86.96±0.21 | 83.40±0.14 | 91.10±0.07 | 82.30±0.46 | 73.78±0.55 |
| $\diamond$PiCO+ | 93.64±0.19 | 93.13±0.26 | 92.18±0.38 | 92.32±0.08 | 92.22±0.01 | 89.95±0.19 | 91.07±0.02 | 89.68±0.01 | 84.08±0.42 |
| $\diamond$IRNet | 93.44±0.21 | 92.57±0.25 | 92.38±0.21 | 92.81±0.19 | 92.18±0.18 | 91.35±0.08 | 91.51±0.05 | 90.76±0.10 | 86.19±0.41 |
| $\diamond$ALIM-Scale | **94.15±0.14** | 93.41±0.04 | 93.28±0.08 | 93.40±0.03 | 92.69±0.11 | 92.01±0.19 | 92.52±0.12 | 90.92±0.10 | 86.51±0.21 |
| $\diamond$ALIM-Onehot | **94.15±0.15** | **94.04±0.16** | **93.77±0.27** | **93.44±0.16** | **93.25±0.08** | **92.42±0.17** | **92.67±0.12** | **91.83±0.08** | **89.80±0.38** |
| $\heartsuit$PiCO+ | 94.58±0.02 | 94.74±0.13 | 94.43±0.19 | 94.02±0.03 | 94.03±0.01 | 92.94±0.24 | 93.56±0.08 | 92.65±0.26 | 88.21±0.37 |
| $\heartsuit$ALIM-Scale | 95.71±0.01 | 95.50±0.08 | 95.35±0.13 | 95.31±0.16 | 94.77±0.07 | 94.36±0.03 | 94.71±0.04 | 93.82±0.13 | 90.63±0.10 |
| $\heartsuit$ALIM-Onehot | **95.83±0.13** | **95.86±0.15** | **95.75±0.19** | **95.52±0.15** | **95.41±0.13** | **94.67±0.21** | **95.19±0.24** | **93.89±0.21** | **92.26±0.29** |

| CIFAR-100 | $q=0.01$ | | | $q=0.03$ | | | $q=0.05$ | | |
|---|---|---|---|---|---|---|---|---|---|
| | $\eta=0.1$ | $\eta=0.2$ | $\eta=0.3$ | $\eta=0.1$ | $\eta=0.2$ | $\eta=0.3$ | $\eta=0.1$ | $\eta=0.2$ | $\eta=0.3$ |
| $\diamond$CC | 53.63±0.46 | 48.84±0.19 | 45.50±0.28 | 51.85±0.18 | 47.48±0.30 | 43.37±0.42 | 50.64±0.40 | 45.87±0.36 | 40.87±0.47 |
| $\diamond$RC | 52.73±1.05 | 48.59±1.04 | 45.77±0.31 | 52.15±0.19 | 48.25±0.38 | 43.92±0.37 | 46.62±0.34 | 45.46±0.21 | 40.31±0.55 |
| $\diamond$LWC | 53.16±0.87 | 48.64±0.33 | 45.51±0.28 | 51.69±0.28 | 47.60±0.44 | 43.39±0.18 | 50.55±0.34 | 45.85±0.28 | 39.83±0.30 |
| $\diamond$LWS | 56.05±0.20 | 50.66±0.59 | 45.71±0.45 | 53.59±0.45 | 48.28±0.44 | 42.20±0.49 | 45.46±0.44 | 39.63±0.80 | 33.60±0.64 |
| $\diamond$PiCO | 68.27±0.08 | 62.24±0.31 | 58.97±0.09 | 67.38±0.09 | 62.01±0.33 | 58.64±0.28 | 67.52±0.43 | 61.52±0.28 | 58.18±0.65 |
| $\diamond$CRDPLL | 68.12±0.13 | 65.32±0.34 | 62.94±0.28 | 67.53±0.07 | 64.29±0.27 | 61.79±0.11 | 67.17±0.04 | 64.11±0.42 | 61.03±0.43 |
| $\diamond$PiCO+ | 71.42±0.25 | 70.22±0.33 | 66.14±0.32 | 70.89±0.13 | 69.03±0.01 | 64.22±0.23 | 69.40±0.12 | 66.67±0.18 | 62.24±0.38 |
| $\diamond$IRNet | 71.17±0.14 | 70.10±0.28 | 68.77±0.28 | 71.01±0.43 | 70.15±0.17 | 68.18±0.30 | 70.73±0.09 | 69.33±0.51 | 68.09±0.12 |
| $\diamond$ALIM-Scale | **74.35±0.18** | **73.36±0.05** | **72.50±0.13** | **74.33±0.11** | **72.60±0.15** | **71.69±0.39** | **73.77±0.05** | **72.39±0.04** | **71.68±0.14** |
| $\diamond$ALIM-Onehot | 72.26±0.23 | 71.98±0.29 | 71.04±0.31 | 71.43±0.21 | 70.79±0.43 | 70.14±0.25 | 72.28±0.28 | 70.60±0.44 | 70.05±0.43 |
| $\heartsuit$PiCO+ | 75.04±0.18 | 74.31±0.02 | 71.79±0.17 | 74.68±0.19 | 73.65±0.23 | 69.97±0.01 | 73.06±0.16 | 71.37±0.16 | 67.56±0.17 |
| $\heartsuit$ALIM-Scale | **77.37±0.32** | **76.81±0.05** | **76.45±0.30** | **77.60±0.18** | **76.63±0.19** | **75.92±0.14** | 76.86±0.23 | **76.44±0.12** | **75.67±0.17** |
| $\heartsuit$ALIM-Onehot | 76.52±0.19 | 76.55±0.24 | 76.09±0.23 | 77.27±0.23 | 76.29±0.41 | 75.29±0.57 | **76.87±0.20** | 75.23±0.42 | 74.49±0.61 |

**Implementation Details**  There are mainly three user-specific parameters in ALIM: $\lambda$, $\lambda_{\mathrm{mix}}$, and $e_0$. Among them, $\lambda$ controls the trade-off between the initial candidate set and model outputs. This paper proposes two selection strategies for $\lambda$, i.e., manually and adaptively adjusted strategies. For the first one, we treat $\lambda$ as a hyper-parameter and select it from $\{0.1, 0.2, 0.3, 0.4, 0.5, 0.7\}$. For the second one, we automatically determine $\lambda$ using the estimated noise level. $\lambda_{\mathrm{mix}}$ controls the trade-off between the PLL loss and the mixup loss, and we set $\lambda_{\mathrm{mix}} = 1.0$ as the default parameter. $e_0$ is the start epoch of ALIM, and we select it from $\{20, 40, 80, 100, 140\}$. Following the standard experimental setup in PLL [17, 25], we split a clean validation set from the training set to determine hyper-parameters. Then, we transform the validation set back to its original PLL form and incorporate it into the training set to accomplish model optimization. To optimize all trainable parameters, we choose the SGD optimizer with a momentum of 0.9 and set the weight decay to 0.001. We set the initial learning rate to 0.01 and adjust it using the cosine scheduler. To eliminate the randomness of the results, we run each experiment three times and report the average result and standard deviation on the test set. All experiments are implemented with PyTorch [27] and carried out with NVIDIA Tesla V100 GPU.

## 3.2 Experimental Results and Discussion

**Main Results**  For a fair comparison, we reproduce all baselines using the same data generation strategy as previous works [12]. In Table 1, we observe that a large portion of improvement is due to mixup training rather than model innovation. To this end, we compare different approaches under the same mixup strategy. Experimental results demonstrate that our method succeeds over all baselines under varying ambiguity levels and noise levels. The main reason lies in two folds. On the one hand, existing PLL methods are mainly designed for clean samples but ignore the presence of noisy samples. Our method can deal with noisy samples by trading off the initial candidate set and model outputs. On the other hand, existing noisy PLL methods generally need to detect noisy samples, but detection errors are unavoidable. These errors can accumulate and continuously affect the training process. Differently, ALIM can deal with this problem by taking advantage of the initial candidate set and restarting correction. These results prove the effectiveness of ALIM in noisy PLL.

**Compatibility of ALIM**  Since ALIM is a plug-in strategy, we integrate it with existing PLL methods and report results in Table 2. We observe that ALIM always brings performance improvement under noisy conditions, verifying the effectiveness and compatibility of our method. Meanwhile,

Table 2: Compatibility of ALIM on different PLL methods.

| PLL | ALIM | CIFAR-10 | | | | | | | | |
|---|---|---|---|---|---|---|---|---|---|---|
| | | $q = 0.1$ | | | $q = 0.3$ | | | $q = 0.5$ | | |
| | | $\eta = 0.1$ | $\eta = 0.2$ | $\eta = 0.3$ | $\eta = 0.1$ | $\eta = 0.2$ | $\eta = 0.3$ | $\eta = 0.1$ | $\eta = 0.2$ | $\eta = 0.3$ |
| ◇RC | × | 80.87±0.30 | 78.22±0.23 | 75.24±0.17 | 79.69±0.37 | 75.69±0.63 | 71.01±0.54 | 72.46±1.51 | 59.72±0.42 | 49.74±0.70 |
| ◇RC | ✓ | 88.81±0.17 | 87.16±0.20 | 85.53±0.05 | 86.21±0.17 | 83.64±0.07 | 79.83±0.43 | 77.40±0.31 | 69.13±0.71 | 56.75±1.59 |
| ◇PiCO | × | 90.78±0.24 | 87.27±0.11 | 84.96±0.12 | 89.71±0.18 | 85.78±0.23 | 82.25±0.32 | 88.11±0.29 | 82.41±0.30 | 68.75±2.62 |
| ◇PiCO | ✓ | 94.15±0.15 | 94.04±0.16 | 93.77±0.27 | 93.44±0.16 | 93.25±0.08 | 92.42±0.17 | 92.67±0.12 | 91.83±0.08 | 89.80±0.38 |
| ◇CRDPLL | × | 93.48±0.17 | 89.13±0.39 | 86.19±0.48 | 92.73±0.19 | 86.96±0.21 | 83.40±0.14 | 91.10±0.07 | 82.30±0.46 | 73.78±0.55 |
| ◇CRDPLL | ✓ | 96.03±0.23 | 95.01±0.32 | 93.36±0.10 | 95.32±0.13 | 93.27±0.29 | 91.20±0.06 | 93.82±0.05 | 90.20±0.04 | 84.24±0.28 |

| PLL | ALIM | CIFAR-100 | | | | | | | | |
|---|---|---|---|---|---|---|---|---|---|---|
| | | $q = 0.01$ | | | $q = 0.03$ | | | $q = 0.05$ | | |
| | | $\eta = 0.1$ | $\eta = 0.2$ | $\eta = 0.3$ | $\eta = 0.1$ | $\eta = 0.2$ | $\eta = 0.3$ | $\eta = 0.1$ | $\eta = 0.2$ | $\eta = 0.3$ |
| ◇RC | × | 52.73±1.05 | 48.59±1.04 | 45.77±0.31 | 52.15±0.19 | 48.25±0.38 | 43.92±0.37 | 46.62±0.34 | 45.46±0.21 | 40.31±0.55 |
| ◇RC | ✓ | 61.46±0.26 | 60.10±0.23 | 55.67±0.28 | 57.43±0.20 | 52.98±0.27 | 48.74±0.37 | 56.40±0.60 | 51.91±0.12 | 46.87±0.74 |
| ◇PiCO | × | 68.27±0.08 | 62.24±0.31 | 58.97±0.09 | 67.38±0.09 | 62.01±0.33 | 58.64±0.28 | 67.52±0.43 | 61.52±0.28 | 58.18±0.65 |
| ◇PiCO | ✓ | 72.26±0.23 | 71.98±0.29 | 71.04±0.31 | 71.43±0.21 | 70.79±0.43 | 70.14±0.25 | 72.28±0.28 | 70.60±0.44 | 70.05±0.43 |
| ◇CRDPLL | × | 68.12±0.13 | 65.32±0.34 | 62.94±0.28 | 67.53±0.07 | 64.29±0.27 | 61.79±0.11 | 67.17±0.04 | 64.11±0.42 | 61.03±0.43 |
| ◇CRDPLL | ✓ | 69.98±0.30 | 68.58±0.16 | 66.90±0.16 | 69.60±0.20 | 67.67±0.22 | 66.15±0.12 | 68.75±0.06 | 67.07±0.29 | 64.69±0.23 |

Table 3: Performance on fine-grained datasets.

| Method | CUB-200 | CIFAR-100H |
|---|---|---|
| | ($q = 0.05, \eta = 0.2$) | ($q = 0.5, \eta = 0.2$) |
| ◇CC | 26.98±1.16 | 34.57±0.99 |
| ◇RC | 44.74±2.47 | 48.03±0.47 |
| ◇LWS | 18.65±2.15 | 22.18±6.12 |
| ◇GCE | 5.13±38.65 | 33.21±2.03 |
| ◇MSE | 20.92±1.20 | 35.20±1.03 |
| ◇CRDPLL | 44.19±0.72 | 53.72±0.21 |
| ◇PiCO | 53.05±2.03 | 59.81±0.25 |
| ♡PiCO+ | 60.65±0.79 | 68.31±0.47 |
| ♡ALIM-Scale | **68.38±0.47** | **73.42±0.18** |
| ♡ALIM-Onehot | 63.91±0.35 | 72.36±0.20 |

Table 4: Performance with severe noise.

| Method | CIFAR-100 | CIFAR-100 |
|---|---|---|
| | ($q = 0.05, \eta = 0.4$) | ($q = 0.05, \eta = 0.5$) |
| ◇RC | 33.64±0.82 | 26.91±0.83 |
| ◇PiCO | 44.17±0.08 | 35.51±1.14 |
| ◇CRDPLL | 57.10±0.24 | 52.10±0.36 |
| ◇ALIM-Scale | **70.37±0.06** | **64.74±0.16** |
| ◇ALIM-Onehot | 68.82±0.37 | 63.39±0.82 |
| ♡PiCO+ | 66.41±0.58 | 60.50±0.99 |
| ♡ALIM-Scale | **74.98±0.16** | **72.26±0.25** |
| ♡ALIM-Onehot | 71.76±0.56 | 69.59±0.62 |

integrating ALIM with PiCO generally achieves better performance under severe noise ($\eta = 0.3$). In noisy label learning, self-training suffers from accumulated errors caused by sample-selection bias [28]. To address this issue, researchers propose co-training [29], which contains multiple branches and cross-updates these branches. Among all PLL methods, PiCO is a natural co-training network with two branches: one for classification and one for clustering [17]. The output of the classification branch guides the model to update the clustering prototypes; the output of the clustering branch affects the update of the pseudo label for classification. These results verify the effectiveness of co-training under noisy conditions. Therefore, we combine ALIM with PiCO in the following experiments.

**Fine-Grained Classification Performance**    Considering that similar categories are more likely to be added to the candidate set, we conduct experiments on more challenging fine-grained datasets, CIFAR-100H [17] and CUB-200 [24]. Different from previous works [12] that generate candidate labels from the entire label space, this section considers the case of generating candidate labels belonging to the same superclass. Experimental results in Table 3 demonstrate that ALIM outperforms existing methods on fine-grained classification tasks, verifying the effectiveness of our method.

**Robustness with Severe Noise**    To demonstrate noise robustness, we conduct experiments under severe noise conditions. In this section, we compare ALIM with the most competitive baselines (RC, PiCO, CRDPLL, and PiCO+) under $\eta \in \{0.4, 0.5\}$. Experimental results in Table 4 show that ALIM succeeds over all baselines under severe noise. Taking the results on $\eta = 0.5$ as an example, our method outperforms the currently advanced approaches by 11.76%. These results demonstrate that our ALIM is more robust to noise than existing PLL and noisy PLL methods.

**Discussion on Normalization Functions**    In this section, we compare the performance of two normalization functions. In Table 1, we observe that Onehot(·) performs slightly better on CIFAR-10 and Scale(·) performs slightly better on CIFAR-100. In Table 3~4, Scale(·) generally achieves better performance in fine-grained datasets and severe noise conditions. The main difference between these normalization functions is that Onehot(·) compresses estimated probabilities into a specific class, while Scale(·) preserves prediction information for all categories. For simple datasets like CIFAR-10, the class with the highest probability is likely to be correct. Therefore, the compression operation can

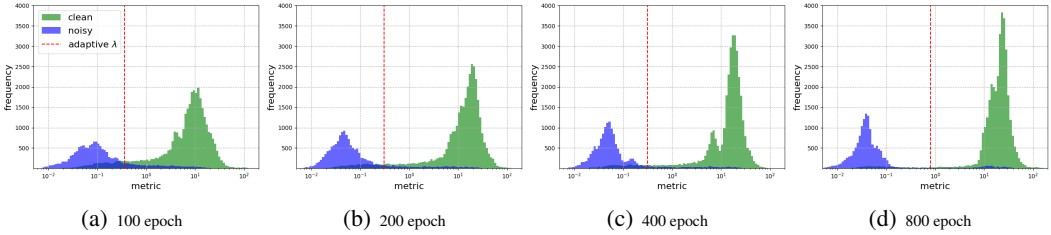

(a) 100 epoch     (b) 200 epoch     (c) 400 epoch     (d) 800 epoch

Figure 2: Distribution of the value in Eq. 6 for clean and noise subsets with increasing training iterations. We conduct experiments on CIFAR-10 ($q = 0.3, \eta = 0.3$) with $e_0 = 80$.

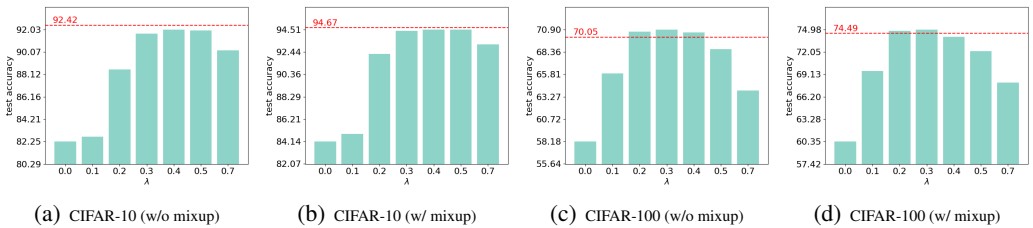

(a) CIFAR-10 (w/o mixup)     (b) CIFAR-10 (w/ mixup)     (c) CIFAR-100 (w/o mixup)     (d) CIFAR-100 (w/ mixup)

Figure 3: Classification performance of manually adjusted $\lambda$ and adaptively adjusted $\lambda$. We conduct experiments on CIFAR-10 ($q = 0.3, \eta = 0.3$) and CIFAR-100 ($q = 0.05, \eta = 0.3$). We mark the results of the adaptively adjusted strategy with red lines.

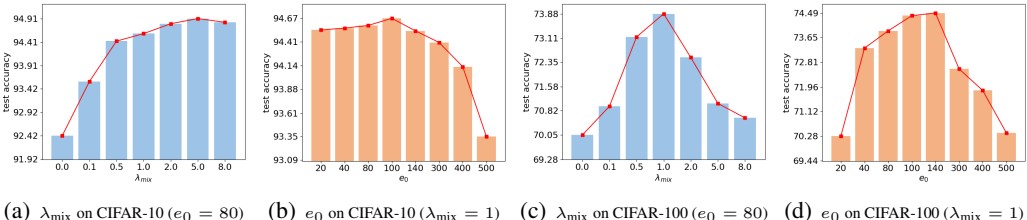

(a) $\lambda_{mix}$ on CIFAR-10 ($e_0 = 80$)     (b) $e_0$ on CIFAR-10 ($\lambda_{mix} = 1$)     (c) $\lambda_{mix}$ on CIFAR-100 ($e_0 = 80$)     (d) $e_0$ on CIFAR-100 ($\lambda_{mix} = 1$)

Figure 4: Parameter sensitivity analysis on CIFAR-10 ($q = 0.3$) and CIFAR-100 ($q = 0.05$) with mixup training. The noise level of these datasets is fixed to $\eta = 0.3$.

reduce the negative impact of other categories and achieve better performance on noisy PLL. For challenging datasets, the class with the highest predicted value has a low probability of being correct. Therefore, this compression operation may lead to severe information loss. More importantly, ALIM consistently outperforms existing methods regardless of the normalization function. Therefore, the main improvement comes from our ALIM rather than normalization functions.

**Rationality of Adaptively Adjusted Strategy**    In this section, we further explain the rationality of our adaptively adjusted strategy. Figure 2 visualizes the distribution of the value in Eq. 6 for clean and noise subsets with increasing training iterations. We observe that this value is an effective indicator for distinguishing clean samples from noisy samples. At the same time, our adaptively adjusted $\lambda$ serves as a suitable boundary for clean and noisy subsets.

**Role of Adaptively Adjusted Strategy**    It is important to select a proper $\lambda$ in ALIM. In this paper, we propose two selection strategies: adaptively and manually adjusted strategies. Figure 3 presents the classification performance of these strategies. Experimental results show that different strategies can achieve similar performance. Therefore, the main advantage of this adaptively adjusted strategy is to reduce manual efforts in hyper-parameter tuning. The performance improvement still comes from ALIM, which exploits a coefficient to control the reliability of the candidate set.

**Parameter Sensitivity Analysis** In this section, we perform parameter sensitivity analysis on two key hyper-parameters: the warm-up epoch $e_0$, and the trade-off between the PLL loss and the mixup loss $\lambda_{\mathrm{mix}}$. In Figure 4, we observe that choosing an appropriate $\lambda_{\mathrm{mix}}$ is important for ALIM, and $\lambda_{\mathrm{mix}} = 1.0$ generally achieves competitive results. Meanwhile, with the gradual increase of $e_0$, the test accuracy first increases and then decreases. This phenomenon indicates that ALIM needs warm-up training. But too large $e_0$ can also cause the model to overfit noise samples.

## 4 Related Work

In PLL, the ground-truth label is concealed in the candidate set. To deal with this task, the core is to disambiguate the candidate labels. In this section, we first introduce two typical disambiguation strategies, i.e., average-based methods and identification-based methods. Then, we briefly review some recent works on noisy PLL.

**Average-based PLL** The most intuitive solution is the average-based approach, which assumes that each candidate label has an equal probability of being the ground-truth label. For example, Hüllermeier et al. [30] utilized k-nearest neighbors for label disambiguation. For each sample, they treated all candidate labels of its neighborhood equally and predicted the ground-truth label through the majority vote. Differently, Cour et al. [31] maximized the average output of candidate labels and non-candidate labels in parametric models. However, these average-based methods can be severely affected by false positive labels [9].

**Identification-based PLL** To address the shortcomings of the above methods, researchers have focused on identification-based methods. Different from average-based methods that treat all candidate labels equally, identification-based methods treat the ground-truth label as a latent variable and maximize its estimated probability by maximum margin [32] or maximum likelihood [5] criteria.

Recently, deep learning has been applied to identification-based methods and achieved promising results. For example, Lv et al. [33] proposed a self-training strategy to disambiguate candidate labels. Feng et al. [18] introduced classifier- and risk-consistent algorithms under the uniform candidate label generation assumption. Wen et al. [25] relaxed this assumption and proposed a family of loss functions for label disambiguation. To learn more discriminative representations, Wang et al. [17] exploited contrastive learning to deal with partially labeled samples. More recently, Wu et al. [26] used consistency regularization on the candidate set and supervised learning on the non-candidate set, achieving promising results under varying ambiguity levels. The above methods rely on a basic assumption that the ground-truth label must be in the candidate set. But this assumption may not be satisfied due to the unprofessional judgment of annotators.

**Noisy PLL** Recently, noisy PLL has attracted increasing attention from researchers due to its more practical setup. Its core challenge is how to deal with noisy samples. For example, Lv et al. [11] utilized noise-tolerant loss functions to avoid overemphasizing noisy samples during training. Lian et al. [12] proposed an iterative refinement network to purify noisy samples and reduce the noise level of the dataset. Wang et al. [13] detected clean samples through a distance-based sample selection mechanism and dealt with noisy samples via a semi-supervised contrastive framework. These noisy PLL methods generally need to detect noisy samples, but detection errors are unavoidable. These errors can accumulate and continuously affect model training. To reduce the negative impact of prediction errors, we propose a novel framework for noisy PLL. Experimental results on multiple datasets demonstrate the effectiveness of our method under noisy conditions.

## 5 Conclusion

This paper introduces a novel framework for noisy PLL called ALIM. To deal with noisy samples, it exploits the weighting mechanism to adjust the reliability of the initial candidate set and model outputs. To verify its effectiveness, we conduct experiments on multiple benchmark datasets under varying ambiguity levels and noise levels. Experimental results demonstrate that our ALIM achieves state-of-the-art classification performance on noisy PLL, especially in severe noise conditions and fine-grained datasets. Meanwhile, ALIM is a low-computation plug-in strategy that can be easily integrated with existing PLL frameworks. Furthermore, we interpret the rationality and effectiveness

of the adaptively adjusted strategy. We also conduct parameter sensitivity analysis and reveal the impact of different hyper-parameters. It is worth noting that this paper leverages a global $\lambda$ to measure the reliability of the candidate set. In the future, we will explore the instance-dependent $\lambda(x)$ during training. At the same time, we will investigate the performance in more experimental settings, such as class-imbalanced conditions.

# 6 Acknowledge

This work is supported by the National Natural Science Foundation of China (NSFC) (No.61831022, No.62276259, No.62201572, No.U21B2010, No.62271083), Beijing Municipal Science&Technology Commission,Administrative Commission of Zhongguancun Science Park No.Z211100004821013, Open Research Projects of Zhejiang Lab (NO. 2021KH0AB06), CCF-Baidu Open Fund (No.OF2022025), the National Natural Science Foundation of China (Grant No. 62106028), Chongqing Overseas Chinese Entrepreneurship and Innovation Support Program, Chongqing Artificial Intelligence Innovation Center, CAAI-Huawei MindSpore Open Fund, and Openl Community (https://openi.pcl.ac.cn).

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

# A Interpretation from Objective Functions

In this section, we provide proofs of the $\text{Onehot}(\cdot)$ normalization function and the $\text{Scale}(\cdot)$ normalization function from the perspective of objective functions.

## A.1 Proof for Onehot Normalization

For $K = 0$, we choose the following objective function during training:

$$\max \sum_{i=1}^{c} w_i \log P_i + M \left( \sum_{i=1}^{c} w_i S_i - 1 \right)$$

$$s.t. \sum_{i}^{c} w_i = 1, w_i \geq 0. \tag{9}$$

Introduce Lagrange multipliers $\delta_i, i \in [1, c]$ and $\gamma$ into Eq. 9, we have:

$$\mathcal{L} = \sum_{i=1}^{c} w_i \log P_i + M \left( \sum_{i=1}^{c} w_i S_i - 1 \right) + \gamma \left( 1 - \sum_{i=1}^{c} w_i \right) + \sum_{i=1}^{c} \delta_i w_i. \tag{10}$$

Combined with the Karush-Kuhn-Tucker (KKT) conditions, the optimal point should satisfy:

$$\log P_i + M S_i - \gamma + \delta_i = 0, \tag{11}$$

$$\sum_{i=1}^{c} w_i = 1, \delta_i \geq 0, w_i \geq 0, \delta_i w_i = 0. \tag{12}$$

Since $S_i \in \{0, 1\}$, we have $M S_i = \log(e^M S_i + (1 - S_i))$. The equivalent equation of Eq. 11 is:

$$\delta_i = \gamma - \log(e^M S_i + (1 - S_i)) P_i. \tag{13}$$

Combined with $\delta_i \geq 0$ in Eq. 12, we have:

$$\gamma \geq \max_i \left( \log(e^M S_i + (1 - S_i)) P_i \right). \tag{14}$$

$\delta_i > 0$ is true if $\gamma > \max_i \left( \log(e^M S_i + (1 - S_i)) P_i \right)$. According to $\delta_i w_i = 0$, we always have $w_i = 0$, which conflicts with $\sum_{i=1}^{c} w_i = 1$. Therefore, we get:

$$\gamma = \max_i \left( \log(e^M S_i + (1 - S_i)) P_i \right). \tag{15}$$

We assume that only one $i_0 \in [1, c]$ reaches the maximum $\gamma$, then we have $w_i = 0, i \in [1, c]/i_0$. Combined with $\sum_{i=1}^{c} w_i = 1$, we get $w_{i_0} = 1$. Therefore, $w(x)$ should satisfy:

$$w(x) = \text{Onehot} \left( \log(e^M S(x) + (1 - S(x))) P(x) \right). \tag{16}$$

We mark $\lambda = e^{-M}$ and convert Eq. 16 to its equivalent version:

$$w(x) = \text{Onehot} \left( (S(x) + \lambda(1 - S(x)) P(x) \right). \tag{17}$$

## A.2 Proof for Scale Normalization

For $K > 0$, $\log w_i$ ensures that $w_i$ must be positive. Therefore, the constraint $w_i \geq 0$ can be excluded. Then, the objective function can be converted to:

$$\max \sum_{i=1}^{c} w_i \log P_i + M \left( \sum_{i=1}^{c} w_i S_i - 1 \right) - K \sum_{i=1}^{c} w_i \log w_i$$

$$s.t. \sum_{i}^{c} w_i = 1. \tag{18}$$

Introduce the Lagrange multiplier $\gamma$ in Eq. 18, we have:

$$\mathcal{L} = \sum_{i=1}^{c} w_i \log P_i + M \left( \sum_{i=1}^{c} w_i S_i - 1 \right) - K \sum_{i=1}^{c} w_i \log w_i + \gamma \left( 1 - \sum_{i}^{c} w_i \right). \qquad (19)$$

Since the optimal point should satisfy $\nabla_w \mathcal{L} = 0$, we have:

$$\log P_i + M S_i - K \left( 1 + \log w_i \right) - \gamma = 0. \qquad (20)$$

Since $S_i \in \{0, 1\}$, we have $M S_i = \log(e^M S_i + (1 - S_i))$. The equivalent equation of Eq. 20 is:

$$\log(e^M S_i + (1 - S_i)) P_i - (K + \gamma) - K \log w_i = 0, \qquad (21)$$

$$w_i^K = \frac{\left( e^M S_i + (1 - S_i) \right) P_i}{e^{K+\gamma}}. \qquad (22)$$

We mark $\lambda = e^{-M}$. Then, we have:

$$w_i = \frac{((S_i + \lambda(1 - S_i)) P_i)^{1/K}}{e^{1+(\gamma-M)/K}}. \qquad (23)$$

Since $\sum_i^c w_i = 1$, we have:

$$\sum_{i=1}^{c} \frac{((S_i + \lambda(1 - S_i)) P_i)^{1/K}}{e^{1+(\gamma-M)/K}} = 1, \qquad (24)$$

$$e^{1+(\gamma-M)/K} = \sum_{i=1}^{c} ((S_i + \lambda (1 - S_i)) P_i)^{1/K}. \qquad (25)$$

Combine Eq. 23 and Eq. 25 and we have:

$$w_i = \frac{((S_i + \lambda(1 - S_i)) P_i)^{1/K}}{\sum_{i=1}^{c} ((S_i + \lambda(1 - S_i)) P_i)^{1/K}}. \qquad (26)$$

Combined with the definition of $\text{Scale}(\cdot)$ in Eq. 4, this equation can be converted to:

$$w(x) = \text{Scale} \left( (S(x) + \lambda(1 - S(x))) P(x) \right). \qquad (27)$$

## B    EM Perspective of ALIM

EM aims to maximize the likelihood of the dataset $\mathcal{D}$:

$$
\begin{aligned}
\max_{\theta} \sum_{x \in \mathcal{D}} \log P(x, S(x); \theta) &= \max_{\theta} \sum_{x \in \mathcal{D}} \log \sum_{i=1}^{c} P(x, S(x), y(x) = i; \theta) \\
&= \max_{\theta} \sum_{x \in \mathcal{D}} \log \sum_{i=1}^{c} w_i(x) \frac{P(x, S(x), y(x) = i; \theta)}{w_i(x)} \\
&\geq \max_{\theta} \sum_{x \in \mathcal{D}} \sum_{i=1}^{c} w_i(x) \log \frac{P(x, S(x), y(x) = i; \theta)}{w_i(x)}, \qquad (28)
\end{aligned}
$$

where $\theta$ is the trainable parameter. The last step of Eq. 28 utilizes Jensen's inequality. Since the $\log(\cdot)$ function is strictly concave, the equal sign takes when $P(x, S(x), y(x) = i; \theta)/w_i(x)$ is some constant $C$, i.e.,

$$w_i(x) = \frac{1}{C} P(x, S(x), y(x) = i; \theta). \qquad (29)$$

Considering that $\sum_{i=1}^{c} w_i(x) = 1$, we can further get:

$$C = \sum_{i=1}^{c} P(x, S(x), y(x) = i; \theta). \tag{30}$$

Then, we have:

$$w_i(x) = \frac{P(x, S(x), y(x) = i; \theta)}{\sum_{i=1}^{c} P(x, S(x), y(x) = i; \theta)} = \frac{P(x, S(x), y(x) = i; \theta)}{P(x, S(x); \theta)} = P(y(x) = i | x, S(x); \theta). \tag{31}$$

In the EM algorithm [16], the E-step aims to calculate $w_i(x)$ and the M-step aims to maximize the lower bound of Eq. 28:

$$\operatorname*{argmax}_{\theta} \sum_{x \in \mathcal{D}} \sum_{i=1}^{c} w_i(x) \log \frac{P(x, S(x), y(x) = i; \theta)}{w_i(x)}$$

$$= \operatorname*{argmax}_{\theta} \sum_{x \in \mathcal{D}} \sum_{i=1}^{c} w_i(x) \log P(x, S(x), y(x) = i; \theta). \tag{32}$$

**E-Step.** In this step, we aim to predict the ground-truth label for each sample:

$$w_i(x) = P(y(x) = i | x, S(x); \theta) = \frac{P(S(x)|y(x) = i, x; \theta) P(y(x) = i | x; \theta)}{P(S(x)|x; \theta)}$$

$$= \frac{P(S(x)|y(x) = i, x; \theta) P(y(x) = i | x; \theta)}{\sum_{i=1}^{c} P(S(x)|y(x) = i, x; \theta) P(y(x) = i | x; \theta)}. \tag{33}$$

According to Assumption 1, we have:

$$P(S(x)|y(x), x) = \begin{cases} \alpha(x), & S_{y(x)}(x) = 1 \\ \beta(x), & S_{y(x)}(x) = 0. \end{cases} \tag{34}$$

It can be converted to:

$$P(S(x)|y(x), x) = \alpha(x) S_{y(x)}(x) + \beta(x) \left(1 - S_{y(x)}(x)\right). \tag{35}$$

Then, we get the equivalent equation of Eq. 33:

$$w_i(x) = \frac{(\alpha(x) S_i(x) + \beta(x)(1 - S_i(x))) P(y(x) = i | x; \theta)}{\sum_{i=1}^{c} (\alpha(x) S_i(x) + \beta(x)(1 - S_i(x))) P(y(x) = i | x; \theta)}. \tag{36}$$

We mark $\lambda(x) = \beta(x)/\alpha(x)$ and $P_i(x) = P(y(x) = i | x; \theta)$. Then, we get:

$$w_i(x) = \frac{(S_i(x) + \lambda(x)(1 - S_i(x))) P_i(x)}{\sum_{i=1}^{c} (S_i(x) + \lambda(x)(1 - S_i(x))) P_i(x)}. \tag{37}$$

It connects traditional PLL and noisy PLL. In traditional PLL, we assume that the ground-truth label must be in the candidate set, i.e., $\beta(x) = 0$. Since $\lambda(x) = \beta(x)/\alpha(x) = 0$, Eq. 37 degenerates to:

$$w_i(x) = \frac{S_i(x) P_i(x)}{\sum_{i=1}^{c} S_i(x) P_i(x)}, \tag{38}$$

which is identical to the classic PLL method, RC [18].

**M-Step.** The objective function of this step is:

$$\operatorname*{argmax}_{\theta} \sum_{x \in \mathcal{D}} \sum_{i=1}^{c} w_i(x) \log P(x, S(x), y(x) = i; \theta)$$

$$= \operatorname*{argmax}_{\theta} \sum_{x \in \mathcal{D}} \sum_{i=1}^{c} w_i(x) \log P(x; \theta) P(y(x) = i | x; \theta) P(S(x) | y(x) = i, x; \theta). \tag{39}$$

Considering that $P(x; \theta) = P(x)$ and $P(S(x)|y(x) = i, x; \theta) = P(S(x)|y(x) = i, x)$, the equivalent version of Eq. 39 is:

$$\underset{\theta}{\text{argmax}} \sum_{x \in \mathcal{D}} \sum_{i=1}^{c} w_i(x) \log P(y(x) = i|x; \theta). \tag{40}$$

Therefore, the essence of the M-step is to minimize the classification loss.

## C  Adaptively Adjusted $\lambda$

Since $\eta$ controls the noise level of the dataset, we have:

$$P(S_{y(x)}(x) = 0) = \eta. \tag{41}$$

After the warm-up training, we assume that the predicted label generated by ALIM $\hat{y}(x) = \arg\max_{1 \leq i \leq c} w(x)$ is accurate, i.e., $\hat{y}(x) = y(x)$. Then we have:

$$P(S_{\hat{y}(x)}(x) = 0) = \eta. \tag{42}$$

To estimate the value of $\lambda$, we first study the equivalent meaning of $S_{\hat{y}(x)}(x) = 0$:

$$\max_{S_i(x)=0} (S_i(x) + \lambda(1 - S_i(x))) P_i(x) \geq \max_{S_i(x)=1} (S_i(x) + \lambda(1 - S_i(x))) P_i(x). \tag{43}$$

We simplify the left and right sides of Eq.43 as follows:

$$\begin{aligned} &\max_{S_i(x)=0} (S_i(x) + \lambda(1 - S_i(x))) P_i(x) \\ &= \max_{S_i(x)=0} \lambda(1 - S_i(x)) P_i(x) \\ &= \max_i \lambda(1 - S_i(x)) P_i(x), \end{aligned} \tag{44}$$

$$\begin{aligned} &\max_{S_i(x)=1} (S_i(x) + \lambda(1 - S_i(x))) P_i(x) \\ &= \max_{S_i(x)=1} S_i(x) P_i(x) \\ &= \max_i S_i(x) P_i(x). \end{aligned} \tag{45}$$

Then, we have:

$$\max_i \lambda(1 - S_i(x)) P_i(x) \geq \max_i S_i(x) P_i(x), \tag{46}$$

$$\lambda \geq \frac{\max_i S_i(x) P_i(x)}{\max_i (1 - S_i(x)) P_i(x)}. \tag{47}$$

Therefore, $P(S_{\hat{y}(x)}(x) = 0) = \eta$ can be converted to:

$$P \left( \lambda \geq \frac{\max_i S_i(x) P_i(x)}{\max_i (1 - S_i(x)) P_i(x)} \right) = \eta. \tag{48}$$

It means that $\lambda$ is the $\eta$-quantile of

$$\left\{ \frac{\max_i S_i(x) P_i(x)}{\max_i (1 - S_i(x)) P_i(x)} \right\}_{x \in \mathcal{D}}. \tag{49}$$

# D Effect of Scaling Factor

Larger $K$ produces smoother pseudo labels. To investigate its impact, we perform a parameter sensitivity analysis in Table 5. We observe that a suitable $K$ is between 0 and 1. Too large $K$ will produce over-smoothed pseudo labels and leads to performance degradation. In this paper, we choose $K = 1$ as the default parameter for the Scale$(\cdot)$ normalization function.

Table 5: Parameter sensitivity analysis on the scaling factor.

| $K$ | CIFAR-10 $(q = 0.3, \eta = 0.3)$ | CIFAR-100 $(q = 0.05, \eta = 0.3)$ |
|---|---|---|
| 0.1 | 94.87±0.08 | 75.29±0.44 |
| 0.2 | 94.72±0.18 | 75.60±0.41 |
| 0.5 | 94.69±0.15 | 75.47±0.03 |
| 1.0 | 94.36±0.03 | 75.67±0.17 |
| 1.5 | 93.04±0.65 | 72.82±0.23 |

