# OpenReview forum: "ALIM: Adjusting Label Importance Mechanism for Noisy Partial Label Learning"
_NeurIPS.cc/2023/Conference — NeurIPS 2023 poster_

### Official Review · Reviewer_q7G6 · 2023-07-04

**Soundness:** 2 fair
**Presentation:** 2 fair
**Contribution:** 3 good
**Rating:** 5
**Confidence:** 3

**Summary:**

This paper addresses the noise partial label learning problem, i.e., the ground-true may also be in the non-candidate set. The proposed method adjusts the importance of the labels by considering the output of the candidate label set model in an integrated manner. The proposed method in this paper can be viewed as a post-processing means for prediction models, and the proposed method achieves by superior performance compared to existing methods on multiple benchmark datasets.

**Strengths:**

1.	The noise PLL issue is important and the authors have a good summary of the unresolved issues in that paper from existing work.
2.	Rich experimental results.


**Weaknesses:**

1. This paper mainly uses the candidate label set and non-candidate label set information to post-process the prediction results, and the innovation is weak.
2. The framework of the method has a stronger assumption that P(x) can also achieve better results in noise partial label learning problem, otherwise it is difficult to inform the subsequent processing. From Figure 1, we can see that for noise partial label learning, P(x) has been able to calculate the confidence of each label well, and the confidence of the true label is higher compared to ALIM.
3. While the way of adjusting label importance proposed in the paper avoids errors when detecting noise directly, the method used in this paper will also assign a certain confidence level to all non-candidate labels, which will also introduce noise.
4. One of the key points of the method proposed in this paper is the setting of the hyper-parameter lambda. An adaptive method is proposed in this paper, but the adaptive parameter is not analyzed in the experimental results to know at what value the setting is reasonable.
5. For Mixup Training, the paper does not explain why it can counteract the effects of noise in PLL.


**Questions:**

1.	In line 91, the authors describe that w(x) should keep very small data, so how can the true labels be extracted from the set of non-candidate labels?
2.	How is the conclusion derived from the different values of K in the paper, which correspond to different normalizations?
3.	What is the principle of Mixup Training in noise partial label learning against noise?

---

> ### Author Rebuttal · Authors · 2023-08-06
>
> # Response to Reviewer q7G6
> We thank the reviewer's appreciation of rich experimental results. We try to address each comment as satisfactorily as possible:
>
> **Q1**: This paper mainly uses the candidate label set and non-candidate label set information to post-process the prediction results, and the innovation is weak.
>
> **A1**: Thanks for your valuable comments, and we would like to present our contribution clearly. This paper proposes ALIM, a plug-in framework for noisy PLL. We perform theoretical analysis from the perspective of objective functions and EM algorithms, demonstrating the feasibility of our method. Experimental results show that ALIM outperforms currently advanced approaches, setting new state-of-the-art records (see Tables 1$\sim$4).
>
> We would like to argue that although our ALIM is easy to implement, it does not mean that our innovation is weak. In contrast, our method has strong motivation and sufficient theoretical guarantees. Such a simple strategy can bring performance improvements to varying PLL approaches, which fully validates its effectiveness.
>
>
> **Q2**: The framework of the method has a stronger assumption that $P(x)$ can also achieve better results in noise partial label learning problem, otherwise it is difficult to inform the subsequent processing. From Figure 1, we can see that for noise partial label learning, $P(x)$ has been able to calculate the confidence of each label well, and the confidence of the true label is higher compared to ALIM.
>
> **A2**: Thanks very much for your valuable comments. In noisy label learning, previous works [1, 2] have demonstrated that when there is a mixture of clean and noisy samples, networks tend to fit the former before the latter. In noisy PLL, researchers [3] also find the same phenomenon. At some intermediate epochs, $P(x)$ can achieve relatively reliable predictions in noisy PLL so that it can be treated as a confidence measure. But if we do not apply ALIM in subsequent training but fully trust $P(x)$, the classification performance will eventually degrade (see Table 2 and Figure 3).
>
> [1] Jiang, Lu, Zhengyuan Zhou, Thomas Leung, Li-Jia Li, and Li Fei-Fei. "Mentornet: Learning data-driven curriculum for very deep neural networks on corrupted labels." In International conference on machine learning, pp. 2304-2313. PMLR, 2018.
>
> [2] Arazo, Eric, Diego Ortego, Paul Albert, Noel O’Connor, and Kevin McGuinness. "Unsupervised label noise modeling and loss correction." In International conference on machine learning, pp. 312-321. PMLR, 2019.
>
> [3] Lian, Zheng, Mingyu Xu, Lan Chen, Licai Sun, Bin Liu, and Jianhua Tao. "Arnet: Automatic refinement network for noisy partial label learning." arXiv preprint arXiv:2211.04774 (2022).
>
>
> **Q3**: While the way of adjusting label importance proposed in the paper avoids errors when detecting noise directly, the method used in this paper will also assign a certain confidence level to all non-candidate labels, which will also introduce noise.
>
> **A3**: Good question! As you pointed out, our method assigns certain confidence to non-candidate labels. For clean samples, this process introduces noise. But for noisy samples, this process allows us to figure out the ground-truth label concealed in the non-candidate set. Therefore, the key is to find the trade-off between them. Experimental results also verify the effectiveness of our strategy in noisy PLL.
>
>
> **Q4**: One of the key points of the method proposed in this paper is the setting of the hyper-parameter $\lambda$. An adaptive method is proposed in this paper, but the adaptive parameter is not analyzed in the experimental results to know at what value the setting is reasonable.
>
> **A4**: Since $\lambda$ changes adaptively during training, we cannot provide its specific value. But according to Figure 2, we prove the rationality of this adaptive parameter. In this figure, we observe that this adaptively adjusted $\lambda$ serves as a suitable boundary for clean and noisy subsets.
>
> **Q5**: For Mixup Training, the paper does not explain why it can counteract the effects of noise in PLL. What is the principle of Mixup Training in noise partial label learning against noise?
>
> **A5**: Our motivation comes from noisy label learning. Since previous works have demonstrated that mixup training is very effective in noisy label learning [1, 2], we wonder whether this strategy can also bring performance improvement for noisy PLL. Experimental results in Table 1 verify our conjecture.
>
> [1] Zhang, Hongyi, Moustapha Cisse, Yann N. Dauphin, and David Lopez-Paz. "mixup: Beyond Empirical Risk Minimization." In International Conference on Learning Representations. 2018.
>
> [2] Li, Junnan, Richard Socher, and Steven CH Hoi. "DivideMix: Learning with Noisy Labels as Semi-supervised Learning." In International Conference on Learning Representations. 2019.
>
>
> **Q6**: In line 91, the authors describe that $w(x)$ should keep very small data, so how can the true labels be extracted from the set of non-candidate labels?
>
> **A6**: We apologize for this unclear description. For clean samples, $w(x)$ should be small at the non-candidate set. But for noisy samples, $w(x)$ should be large at the non-candidate set. Therefore, the key is to find the trade-off between them. In this paper, we do not strictly restrict this objective but use a penalty factor $M$ (see Eq. 5). Larger $M$ means that we have tighter constraints on $\left(\sum_{i=1}^cw_iS_i-1\right)$.
>
> **Q7**: How is the conclusion derived from the different values of $K$ in the paper, which correspond to different normalizations?
>
> **A7**: We thank the reviewer’s valuable comments. According to our proof in Appendix A, the penalty factor $K$ is different for two normalization functions: $K=0$ for $\text{Onehot}(\cdot)$ and $K>0$ for $\text{Scale}(\cdot)$. The main reason is that, for $K>0$, $\log w_i$ ensures that $w_i$ must be positive. Therefore, we can exclude the constraint $w_i \geq 0$ in Eq. 5.

---

> > ### Comment · Reviewer_q7G6 · 2023-08-15
> >
> > Thanks for the response. The replies resolved some technical details and I'd like to raise my score to 5.

---

### Official Review · Reviewer_BGEz · 2023-07-05

**Soundness:** 3 good
**Presentation:** 3 good
**Contribution:** 3 good
**Rating:** 7
**Confidence:** 3

**Summary:**

This paper focuses on a relaxed version of partial label learning, called noisy PLL. Different from PLL, noisy PLL allows the ground-truth label may not be in the candidate set. To deal with this problem, this paper proposes a framework called ALIM, which is a plug-in strategy with theoretical guarantees. Experimental results on CIFAR-10, CIFAR-100, and fine-grained datasets prove the effectiveness of ALIM on noisy PLL. This paper is well-written and easy to follow.

**Strengths:**

1.	This paper presents a plug-and-play framework for the challenging noisy PLL task.
2.	This paper conducts theoretical analysis from objective functions and EM algorithms.
3.	ALIM shows competitive performance compared with the currently advanced strategies.


**Weaknesses:**

$Scale(\cdot)$ normalization function introduces a scaling factor $K$. But in the following experiments, this paper does not specify the selection process of this factor. Please explain its impact on the performance and how it was chosen.

**Questions:**

Can the authors explain the impact of the scaling factor $K$ on the performance and how it was chosen?

**Limitations:**

Please refer to the weakness and question.

---

> ### Author Rebuttal · Authors · 2023-08-06
>
>
> # Response to Reviewer BGEz
> We thank the reviewer's appreciation of the writing and sufficient experimental results. We try to address your concerns as follows:
>
> **Q1**: $\text{Scale}(\cdot)$ normalization function introduces a scaling factor $K$. But in the following experiments, this paper does not specify the selection process of this factor. Please explain its impact on the performance and how it was chosen.
>
> **A1**: Thanks for your valuable comments. The scaling factor $K$ affects the smoothness of pseudo labels. Larger $K$ produces smoother pseudo labels. To investigate its impact, we perform a parameter sensitivity analysis in Table 1. We observe that a suitable $K$ is between 0 and 1. Too large $K$ will produce over-smoothed pseudo labels and leads to performance degradation. In this paper, we choose $K=1$ as the default parameter for $\text{Scale}(\cdot)$ normalization function.
>
> | $K$  | CIFAR-10  ($q=0.3, \eta=0.3$) | CIFAR-100 ($q=0.05, \eta=0.3$) |
> | ---- | ----------------------------- | ------------------------------ |
> | 0.1  |   94.87±0.08                        |          75.29±0.44                      |
> |  0.2     |    94.72±0.18                           |     75.60±0.41                           |
> |  0.5    |       94.69±0.15                        |         75.47±0.03                       |
> |1.0|94.36±0.03|75.67±0.17|
> |1.5|93.04±0.65|72.82±0.23|

---

> > ### Comment · Reviewer_BGEz · 2023-08-15
> >
> > Thanks for the authors' response. I will keep my score.

---

### Official Review · Reviewer_7LXj · 2023-07-05

**Soundness:** 2 fair
**Presentation:** 2 fair
**Contribution:** 3 good
**Rating:** 5
**Confidence:** 4

**Summary:**

The paper considers noisy partial label learning when the provided label is not in the candidate label set. The paper proposes "Adjusting Label Importance Mechanism" to trade off the initial candidate set and model outputs. Some theoretical analysis is done and good empirical results are shown against selected baselines.

**Strengths:**

1. The considered problem (i.e. the provided label is not in the candidate label set) might be a relatively new problem and is less considered in existing work.

2. Some theoretical analysis is done.

3. A few tricks are proposed to make the method more effective.

4. Good experiment results are shown against baselines.

**Weaknesses:**

1. Writing can be improved. Currently, it seems the manuscript is prepared in a hurry. For example, there are errors when referring to appendices: “Appendix ??”. Overall, I had a hard time following.

2. The core contribution is not clear. The motivation in Section 2.2 is hard to follow. From Section 2.2, it seems the paper is trying to consider the case when the ground-truth label is not in the provided candidate set of labels. It should be made clear why this is an important problem and how the proposed method addresses the problem. Also, there are concerns on the core technique. see Q1 and Q4.

3. The theoretical analysis is not clear see Q2.

**Questions:**

1. When the label is not in the candidate set, it’s also likely that it's not in 1-S(x). Actually, not being in the label space might even be the main reason that the label is not in the candidate set. How to handle this case? It seems there is an assumption that the label is for sure in the given label space, and it might not be in the provided candidate which is a subset of the label space. This should be made clear.

2. In Equation 5, why does M correspond to the lambda in equation 2? In section 2.4.2, it is also unclear what is the connection to EM.

3. Abstract mentions theoretical guarantees, what are the guarantees? It is not mentioned in the paper anywhere.

4. From the theoretical analysis, it seems there is an equivalence between the proposed method to an (existing?) objective function. Then, why would one need the proposed method?

**Limitations:**

Limitations are not discussed. Authors should discuss when the proposed method doesn't work.

---

> ### Author Rebuttal · Authors · 2023-08-06
>
> # Response to Reviewer 7LXj
> We sincerely thank the reviewer for your detailed comments and suggestions. We try to address each comment as satisfactorily as possible:
>
> **Q1**: Writing can be improved. Currently, it seems the manuscript is prepared in a hurry. For example, there are errors when referring to appendices: “Appendix ??”. Overall, I had a hard time following.
>
> **A1**: We apologize for these typos. In Lines 95, 107, and 126, these symbols indicate Appendix A, Appendix B, and Appendix C, respectively. We will correct these typos in the revised paper. Meanwhile, we are trying our best to improve the quality of writing.
>
> **Q2**: The core contribution is not clear. The motivation in Section 2.2 is hard to follow. From Section 2.2, it seems the paper is trying to consider the case when the ground-truth label is not in the provided candidate set of labels. It should be made clear why this is an important problem and how the proposed method addresses the problem.
>
> **A2**: Thank you very much for your valuable comments. In the revised version, we clarify the importance of this task and present our approach more clearly.
>
> *Importance of the task*: PLL is a typical type of weakly supervised learning. Due to the low annotation cost of partially labeled samples, PLL has attracted increasing attention from researchers and has been applied to many areas. The basic assumption of traditional PLL is that the ground-truth label must be in the candidate label set. However, this assumption may not be satisfied due to the unprofessional judgment of annotators. Recently, noisy PLL has attracted increasing attention from researchers due to its more practical setup. In noisy PLL, the ground-truth label may not be in the candidate label set. *Reviewer q7G6 also highlights the importance of noisy PLL.*
>
> *Methodology*: In this paper, we propose a novel framework for noisy PLL with theoretical guarantees, called ALIM. It exploits the weighting mechanism to adjust the reliability of the initial candidate set and model outputs. To reduce manual efforts in hyper-parameter tuning, we propose an adaptive strategy to determine the weighting coefficient. To further improve noise tolerance, we equip ALIM with mixup training. We also perform theoretical analysis from the perspective of objective functions and EM algorithms and prove the feasibility of our method.
>
> **Q3**: When the label is not in the candidate set, it’s also likely that it's not in $1-S(x)$. Actually, not being in the label space might even be the main reason that the label is not in the candidate set. How to handle this case? It seems there is an assumption that the label is for sure in the given label space, and it might not be in the provided candidate which is a subset of the label space. This should be made clear.
>
> **A3**: Good question! Current works on noisy PLL [1, 2, 3] assume that the ground-truth label must be in the given label space. For a fair comparison, we also conduct experiments under this assumption. Your comment points a new direction for noisy PLL. In the future, we will try to combine noisy PLL with out-of-distribution (OOD) detection strategies to address this challenging task.
>
> [1] Lv, Jiaqi, Biao Liu, Lei Feng, Ning Xu, Miao Xu, Bo An, Gang Niu, Xin Geng, and Masashi Sugiyama. "On the robustness of average losses for partial-label learning." IEEE Transactions on Pattern Analysis and Machine Intelligence (2023).
>
> [2] Lian, Zheng, Mingyu Xu, Lan Chen, Licai Sun, Bin Liu, and Jianhua Tao. "Arnet: Automatic refinement network for noisy partial label learning." arXiv preprint arXiv:2211.04774 (2022).
>
> [3] Wang, Haobo, Ruixuan Xiao, Yixuan Li, Lei Feng, Gang Niu, Gang Chen, and Junbo Zhao. "PiCO+: Contrastive Label Disambiguation for Robust Partial Label Learning." arXiv preprint arXiv:2201.08984 (2022).
>
>
> **Q4**: In Equation 5, why does $M$ correspond to the lambda in equation 2? In section 2.4.2, it is also unclear what is the connection to EM.
>
> **A4**: We thank the reviewer’s comments. We have provided proofs in Section 2.4 and Appendix.
>
> In Appendix A, we prove the relationship between $\lambda$ and $M$, i.e., $\lambda = e^{-M}$.
>
> In Appendix B, we explain ALIM from an EM perspective. We prove that the E-step aims to predict the ground-truth label for each sample and the M-step aims to minimize the classification loss.
>
>
> **Q5**: Abstract mentions theoretical guarantees, what are the guarantees? It is not mentioned in the paper anywhere.
>
> **A5**: We have provided theoretical guarantees in Section 2.4 and Appendix. To verify the feasibility of ALIM, we conduct theoretical analysis from the perspective of objective functions (see Appendix A) and EM algorithms (see Appendix B).
>
>
> **Q6**: From the theoretical analysis, it seems there is an equivalence between the proposed method to an (existing?) objective function. Then, why would one need the proposed method?
>
> **A6**: In this paper, we propose a novel framework for noisy PLL with theoretical guarantees. The necessity of our method can be illustrated in two aspects:
>
> (1) Some influential works in PLL [1, 2] are also derived from the loss function. The core lies in how to properly design the loss function and make the assumption to obtain an easy-to-implement formula. This process requires a lot of efforts.
>
> (2) Noisy PLL is an emerging direction of machine learning. This objective function is specifically designed for noisy PLL, which does not exist in the previous work.
>
> [1] Feng, Lei, Jiaqi Lv, Bo Han, Miao Xu, Gang Niu, Xin Geng, Bo An, and Masashi Sugiyama. "Provably consistent partial-label learning." Advances in neural information processing systems 33 (2020): 10948-10960.
>
> [2] Wen, Hongwei, Jingyi Cui, Hanyuan Hang, Jiabin Liu, Yisen Wang, and Zhouchen Lin. "Leveraged weighted loss for partial label learning." In International Conference on Machine Learning, pp. 11091-11100. PMLR, 2021.

---

> > ### Comment · Reviewer_7LXj · 2023-08-15
> >
> > I thank the authors for answering my questions. My concerns are addressed and I'm increasing my rating.

---

### Official Review · Reviewer_7tpZ · 2023-07-07

**Soundness:** 3 good
**Presentation:** 3 good
**Contribution:** 3 good
**Rating:** 7
**Confidence:** 3

**Summary:**

This paper focused on the problem of noisy partial label learning (noisy PLL), which means the candidate label set in the partial label learning does not always contain the ground-truth label. Compared with the existing studies which attempted to detect noisy samples yet lead to the accumulation of detection errors, this work proposed to reduce the impact of the detection errors by training off the initial candidate set and model outputs. The proposed method, named "Adjusting Label Importance Mechanism (ALIM)", can adaptively adjust the weighting coefficient between the initial candidate set and the model predictions (i.e., pseudo-labels output by the model). The theoretical analysis showed that the proposed method can be well interpreted from two perspectives of heuristic design and classic Expectation-maximization (EM) algorithm. Combined with Mixup training strategies, the proposed approach can be further plugged into the existing noisy PPL baselines and bring consistent improvements. Extensive experiments demonstrated the effectiveness of the proposed framework.

**Strengths:**

- The motivation is clear. The analysis of the issues of the existing methods motivated the proposed method in a straightforward manner.
-  The proposed objective can be interpreted under the EM framework, which provided a convincing mathematical explanation.
- The proposed method, ALIM, is also compatible with other related approaches. It can work as a plug-in in the existing noisy PLL algorithms, which enhanced the flexibility of ALIM.
- The experimental part conducted empirical comparisons/ablation studies with respect to different factors, e.g., the different noise levels and ambiguity levels, and the different choices of the key components within the proposed framework.

**Weaknesses:**

- Some technical details were not clear to the reviewers. See the next part for more details.

**Questions:**

- In Line 140, what is the definition of $\mathcal{L}_{pll}$? It suddenly appeared here without any pre-defined formation. Which kind of PLL loss did it indicate, the objective in Eq.(5)?
- I recommend the authors reformulate the optimization objective in Eq. (5) as a $\min$ form, which will be better compatible with the form of the loss function.
- Compared to the results reported on the CIFAR10 dataset, the reviewer found that ALIM-Scale consistently performed better than ALIM-Onehot on other datasets like CIFAR100 and CUB200. Is it a necessity that scaling performs better than one shot on datasets with larger class numbers, or a coincidence? Could the author provide more analyses and explanations about this point?
- Some potential typos:
    - Line 95, Appendix ??
    - Line 107, Appendix ??
    - Line 126, Appendix ??

---

> ### Author Rebuttal · Authors · 2023-08-06
>
> # Response to Reviewer 7tpZ
> We thank the reviewer's appreciation of clear writing and rich experimental results. We try to address your concerns as follows:
>
> **Q1**: In Line 140, what is the definition of $L_{\text{pll}}$? It suddenly appeared here without any pre-defined formation. Which kind of PLL loss did it indicate, the objective in Eq. 5?
>
> **A1**: We apologize for this unclear expression. ALIM is a plug-in strategy that can be integrated with existing PLL approaches. In this paper, $L_{\text{pll}}$ means the loss function of PLL approaches. Since different PLL methods use distinct formulations of $L_{\text{pll}}$, we do not provide the specific formulation of $L_{\text{pll}}$. We will clarify it in the revised paper.
>
>
> **Q2**: I recommend the authors reformulate the optimization objective in Eq. (5) as a min form, which will be better compatible with the form of the loss function.
>
> **A2**: Thank you very much for your valuable comments. In the revised version, we reformulate the objective function as follows:
> \begin{align}
> &\min\ -\sum_{i=1}^cw_i\log P_i - M \left(\sum_{i=1}^cw_iS_i-1\right) + K \sum_{i=1}^cw_i \log w_i \nonumber\\
> &s.t. \sum_{i}^c w_i =1, w_i \geq 0, \nonumber
> \end{align}
>
>
> **Q3**: Compared to the results reported on the CIFAR10 dataset, the reviewer found that ALIM-Scale consistently performed better than ALIM-Onehot on other datasets like CIFAR100 and CUB200. Is it a necessity that scaling performs better than one shot on datasets with larger class numbers, or a coincidence? Could the author provide more analyses and explanations about this point?
>
> **A3**: Good question! The main difference between these normalization functions is that $\text{Onehot}(\cdot)$ compresses estimated probabilities into a specific class, while $\text{Scale}(\cdot)$ preserves prediction information for all categories. For simple datasets like CIFAR-10, the class with the highest probability is likely to be correct. Therefore, the compression operation can reduce the negative impact of other categories and achieve better performance on noisy PLL. For challenging datasets, the class with the highest predicted value has a low probability of being correct. Therefore, this compression operation may lead to severe information loss. More importantly, ALIM consistently outperforms existing methods regardless of the normalization function (see Table 1). Therefore, the main improvement comes from our ALIM rather than normalization functions. We will add these discussions to the revised paper.
>
>
> **Q4**: Some potential typos.
>
> **A4**: We apologize for these typos. In Lines 95, 107, and 126, these symbols indicate Appendix A, Appendix B, and Appendix C, respectively. We will correct these typos in the revised version.

---

> > ### Comment · Reviewer_7tpZ · 2023-08-15
> > **Response to Author Rebuttal**
> >
> > Thanks for the effort from the authors in answering my questions. My concerns have been addressed and I accordingly increase my rating to 7.

---

### Author Rebuttal · Authors · 2023-08-06


# General Response
Dear Reviewers, Area Chairs, and Program Chairs:

We would like to express our gratitude to all reviewers for taking their valuable time to review our paper. We sincerely appreciate the reviewers for noting that our motivation is "clear" (7tpZ), the mathematical explanation is "convincing" (7tpZ), and the proposed method is "flexible" (7tpZ) and "effective" (7LXj). We also thank the reviewers for noting that our paper is "well-written and easy to follow" (BGEz), the noisy PLL problem is "important" (q7G6), and our experimental results are "good" (7LXj), "competitive" (BGEz), and "rich" (q7G6). Meanwhile, we appreciate the reviewers for pointing out the shortcomings. Your valuable comments help us improve this paper. We try to address each comment as satisfactorily as possible. Please find the responses to each reviewer’s comments below.

**Motivation and Main Contribution**

Noisy PLL is an important branch of weakly supervised learning. Unlike PLL where the ground-truth label must be concealed in the candidate label set, noisy PLL relaxes this constraint and allows the ground-truth label may not be in the candidate label set. Most of the existing works attempt to detect noisy samples and estimate the ground-truth label for each noisy sample. However, detection errors are unavoidable. These errors can accumulate during training and continuously affect model optimization. To alleviate this issue, we propose a *plug-in* framework for noisy PLL with *theoretical guarantees*. Due to its compatibility, it can be *easily integrated with existing PLL methods* and achieves performance improvement under noisy conditions. Its *unprecedented success* at varying ambiguity levels and noise levels has paved a new way for noisy PLL and can inspire more relevant research.

**Summary of Reviews**

Reviewer 7tpZ and BGEz vote for acceptance, while Reviewer 7LXj and q7G6 give negative comments. Therefore, we try to address the main concerns of Reviewer 7LXj and q7G6:

Reviewer 7LXj gives negative comments on the writing and contribution. We deeply appreciate the reviewer's valuable comments. We are trying our best to improve the quality of writing. It is also worth noting that the evaluation of writing can be subjective, varying from individual to individual. For example, Reviewer 7tpZ and BGEz give high scores for our presentation, soundness, and contribution. In the revised version, we clarify the importance of this task and present our approach more clearly. This paper focuses on an important task in machine learning, noisy PLL. To deal with this problem, we propose a plug-in framework with theoretical guarantees. Due to its compatibility, it can be easily integrated with existing PLL methods. Experimental results on multiple datasets demonstrate its effectiveness.

Reviewer q7G6 raises concerns about our innovation because ALIM looks like a post-processing strategy for prediction results. We would like to argue that although our method is easy to implement, it does not mean that our innovation is weak. In contrast, ALIM has strong motivation and sufficient theoretical guarantees. Such a simple strategy can bring performance improvements to varying PLL approaches, which fully validates its effectiveness. Meanwhile, the reviewer raises concerns about some modules. In the response, we try to address these concerns as satisfactory as possible.

We kindly ask the reviewers to take the above clarifications into account when considering score adjustments. We welcome any further discussion with the reviewers.

Best regards,

Paper4978 Authors

---

### Decision · Program_Chairs · 2023-09-21

**Decision:**

Accept (poster)

**Comment:**

This paper proposes Adjusting Label Importance Mechanism (ALIM), a method for noisy partial label learning. The setting is noisy in the sense that the training set of labels for an example might not actually contain the true label. ALIM takes a pseudolabeling approach, in which the predictions of the model being trained are weighed against the training partial labels. Essentially, confident predictions of the model can overrule noisy labels. Experiments show that ALIM can outperform state of the art methods on CIFAR 10, CIFAR 100, CUB 200 and CIFAR 100H.

In general, the reviewers liked the extensive experimental evaluation and the fact that ALIM can be plugged into many existing approaches. Some of the reviewers had doubts about the formulation and explanation of ALIM that were resolved during the discussion period. The authors are encouraged to add these clarifications to the final version.

One weakness that the reviewers identified is ambiguity in the claim of theoretical "guarantees." The theoretical analysis given is an interpretation, not a guarantee. No guarantee is stated about the performance of the algorithm in any sense. The analysis only shows that the behavior of ALIM can be interpreted in terms of well known frameworks like EM. The authors are strongly encouraged to clarify this in their final version.